# A GNSS/INS/LiDAR Integration Scheme for UAV-Based Navigation in GNSS-Challenging Environments

**DOI:** 10.3390/s22249908

**Published:** 2022-12-16

**Authors:** Ahmed Elamin, Nader Abdelaziz, Ahmed El-Rabbany

**Affiliations:** 1Department of Civil Engineering, Toronto Metropolitan University, Toronto, ON M5B 2K3, Canada; 2Department of Civil Engineering, Faculty of Engineering, Zagazig University, Zagazig 44519, Egypt; 3Department of Civil Engineering, Tanta University, Tanta 31527, Egypt

**Keywords:** UAV, optimized LOAM SLAM, INS/LiDAR SLAM integration, integrated navigation system

## Abstract

Unmanned aerial vehicle (UAV) navigation has recently been the focus of many studies. The most challenging aspect of UAV navigation is maintaining accurate and reliable pose estimation. In outdoor environments, global navigation satellite systems (GNSS) are typically used for UAV localization. However, relying solely on GNSS might pose safety risks in the event of receiver malfunction or antenna installation error. In this research, an unmanned aerial system (UAS) employing the Applanix APX15 GNSS/IMU board, a Velodyne Puck LiDAR sensor, and a Sony a7R II high-resolution camera was used to collect data for the purpose of developing a multi-sensor integration system. Unfortunately, due to a malfunctioning GNSS antenna, there were numerous prolonged GNSS signal outages. As a result, the GNSS/INS processing failed after obtaining an error that exceeded 25 km. To resolve this issue and to recover the precise trajectory of the UAV, a GNSS/INS/LiDAR integrated navigation system was developed. The LiDAR data were first processed using the optimized LOAM SLAM algorithm, which yielded the position and orientation estimates. Pix4D Mapper software was then used to process the camera images in the presence of ground control points (GCPs), which resulted in the precise camera positions and orientations that served as ground truth. All sensor data were timestamped by GPS, and all datasets were sampled at 10 Hz to match those of the LiDAR scans. Two case studies were considered, namely complete GNSS outage and assistance from GNSS PPP solution. In comparison to the complete GNSS outage, the results for the second case study were significantly improved. The improvement is described in terms of RMSE reductions of approximately 51% and 78% for the horizontal and vertical directions, respectively. Additionally, the RMSE of the roll and yaw angles was reduced by 13% and 30%, respectively. However, the RMSE of the pitch angle was increased by about 13%.

## 1. Introduction

In unmanned air vehicle (UAV) navigation, the main challenge is to maintain accurate and robust estimation of the UAV position and orientation. Typically, global navigation satellite systems (GNSS) are used for UAV localization in outdoor environments. However, GNSS signals may suffer from degradation or outages due to a signal blockage or a malfunction, which leads to a vulnerable system if it relies exclusively on GNSS. Therefore, the navigation system must possess redundancy to ensure reliability and availability. This emphasizes the need for multi-sensor integration to achieve reliable autonomous navigation [1,2,3]. In addition, recent developments in sensor technology have increased the payload capabilities of UAVs to include multiple sensors, such as cameras and light detection and ranging (LiDAR).

An inertial navigation system (INS) uses an inertial measurement unit (IMU) to measure the rotation rate and acceleration, which are used to obtain the UAV pose (i.e., position and the attitude) [4,5,6]. However, low-cost MEMS-based IMU measurements suffer from significant drift over time. Consequently, another sensor is required to constrain the drift, such as GNSS [7]. GNSS-aided INS is the most common navigation system [8,9]. Typically, Kalman filter is used to fuse the observations of the GNSS and IMU [10]. For example, in loosely coupled (LC) integration, the INS mechanization provides the position of the vehicle at a high frequency. GNSS updates are fed into the system at a lower frequency to minimize the INS solution drift. Due to the closed-error scheme integration, the INS can provide long-term stability with high accuracy while experiencing occasional short-period GNSS signal outages [11]. However, if the GNSS signal outage persists for a long period, the system will be reliant on the performance of the INS, which has a diverging error nature, especially when a low-cost IMU is used [12,13,14,15,16].

An alternative approach is to use additional onboard sensors, such as cameras and LiDAR sensors, in order to sustain accurate pose estimation of the UAV during GNSS outages. Vision-based navigation systems are mainly integrating visual sensors with the inertial unit through simultaneous localization and mapping (SLAM) techniques, such as ORB-SLAM2 [17] and VINS-Mono [18]. The basic idea of SLAM algorithms is to keep track of the sensor location while simultaneously constructing a map of the surrounding environment [19]. Nevertheless, the camera sensors are susceptible to illumination changes and require detecting sufficient features in the surrounding environment. Consequently, these systems will not operate properly in light-changing situations or featureless areas. 

On the other hand, LiDAR sensors are widely used for navigation using SLAM techniques. Many studies have investigated the use of LiDAR for UAV navigation [20,21,22]. In 2014, the LiDAR odometry and mapping (LOAM) algorithm was presented as one of the leading algorithms [23], which is ranked third on the KITTI benchmark [24]. Subsequently, other variants and modified versions of LOAM were presented, including, A-LOAM [25], LeGO-LOAM [26], optimized LOAM SLAM by kitware [27], R-LOAM [28], and F-LOAM [29]. Improving the processing time of the LOAM algorithm was the common theme among these SLAM algorithms. For UAV navigation, LiDAR is typically used for mapping applications. However, it can potentially be used for navigation to complement onboard positioning sensors. 

GNSS/INS/LiDAR SLAM integration has been investigated by a number of researchers. In [30], a three-dimensional reduced-inertial-sensor system (3D-RISS) was integrated with LiDAR odometry and GNSS through an extended Kalman filter (EKF). The position and azimuth estimated using LiDAR odometry were used to update the 3D-RISS mechanization. The proposed approach was tested for various driving scenarios in Kingston and downtown Toronto, Canada. In comparison with the INS solution, the RMSE of the position in the horizontal direction was reduced by 64%. In [31], graph optimization LiDAR SLAM was integrated with GNSS/INS, where the GNSS/INS results were matched with the relative pose of a 3D probability map. The integrated scheme was tested during a one-minute outage of the GNSS signal, which resulted in reductions in RMSE by 80% approximately. In [32], a similar approach was proposed; however, an odometer was added to the integrated scheme. The system was tested during a 2-min GNSS outage. The integration results showed a reduction in the north, east, and height directions RMSE by 62.8%, 72.3%, and 52.1%, respectively, in comparison with the GNSS/INS integrated system counterpart. In [33], an INS/LiDAR SLAM terrestrial navigation system was developed. The data source in that study was the raw KITTI dataset for both residential and highway drives [24]. Three case studies with different driving scenarios were considered in order to test the robustness of the developed integrated navigation system. The proposed model outperformed a number of the literature LiDAR SLAM algorithms. A similar approach to the one adopted in [33] will be used in this current study, albeit for an UAV navigation. 

In this study, an integrated UAV-based GNSS/INS/LiDAR SLAM navigation system is developed. The proposed approach is developed to overcome frequent and prolonged GNSS outages resulting from an unexpected antenna malfunction during data collection. The data has been collected using a DJI Matrice-600 UAV equipped with the Velodyne PUCK LiDAR sensor (previously VLP-16), a Sony α7R II camera, and the Applanix APX-15 UAV GNSS/INS. The system implements a loosely-coupled integration that takes advantage of the LiDAR SLAM solution (latitude, longitude, altitude, pitch, roll, and yaw) to update the INS solution in a closed-error loop scheme using an EKF. The remainder of this paper is organized as follows: Section 2 presents the UAV data collection platform, data acquisition, and data preprocessing. The system architecture and models are explained in Section 3. Section 4 discusses the experimental results, and finally conclusions are presented in Section 5.

## 2. Data Acquisition and Preprocessing

### 2.1. Data Acquisition

In this study, a UAV was used as the platform for data acquisition. The UAV consisted of a DJI Matrice 600 Pro carrying a Sony α7R II RGB camera, a Velodyne PUCK LiDAR sensor, and the Applanix APX-15 UAV GNSS/INS board. Figure 1 shows the system components and how the various sensors are rigidly fixed within the UAV platform. The Sony α7R II camera has a 42.4 MP spatial resolution with a 7952 × 5304 image size [34]. The images were acquired based on distance, rather than time interval, with an average frame rate of 0.5 Hz, and the number of captured images was 110 images. The Velodyne PUCK, on the other hand, consists of 16 laser beams with a 360° horizontal field of view (FOV) and a 30° vertical field of view. The PUCK maximum range measurement is 100 m with a 3 cm range accuracy [35]. The total number of acquired points was approximately 14 million points at a 10 Hz frame rate. The Applanix APX-15 UAV GNSS/INS unit was used in this study due to its low weight, compact size, and precise positioning and orientation information [36]. The GNSS data was collected at a 10 Hz, while the IMU data was collected at a 200 Hz frame rate. In order to cover the area of interest, the flight was performed at approximately 25 m above the ground, acquiring the data in 5 parallel flight lines. Figure 2 shows a top view of the flight trajectory over the area of interest. 

### 2.2. Data Preprocessing

Applanix POSPac UAV software [37] was used to process the collected GNSS/IMU data in differential mode to obtain the precise drone trajectory. Unfortunately, the software crashed after obtaining an error that exceeded 25 km. Investigating the reason revealed the fact that there were numerous prolonged GNSS signal outages due to a malfunction, as shown in Figure 3. The software estimation filter could not handle the large drift of low-cost IMU. An attempt was made to process the GNSS-only data in differential mode, which also failed. The GNSS data was then processed in precise point positioning (PPP) mode using the Canadian Spatial Reference System PPP (CSRS-PPP) service by Natural Resources Canada (NRCan) [38], which was successful, but the obtained results were at dm-m level accuracy with many gaps. The PPP results indicated that the drone altitude was not uniform (with variation in the range of 5 m), which might have been caused by a mass imbalance in the UAS. Partial or complete reset of the ambiguity parameters were performed frequently, which indicates that the PPP results were essentially code-based, i.e., with degraded accuracy.

As a result of POSPac crash and the unavailability of precise GNSS/INS solution, the camera images had to be processed using ground control points (GCPs) to enable a precise UAV trajectory, which can then serve as ground truth. Twelve checkerboard targets were deployed at the study site, as shown in Figure 4, The precise coordinates (centimeter-level) of the targets were determined using a dual-frequency SOKKIA GCX2 GPS receiver in the real-time kinematic (RTK) mode. The RGB images were processed with the four corner GCPs (green boxes) using Pix4D mapper software [39] to estimate the camera poses. The other eight points (blue boxes) were used as check points. Figure 5 shows the estimated poses of the camera using the Pix4D mapper software. Table 1 shows the absolute camera position and orientation error statistics. 

Finally, the LiDAR data were processed using the optimized LOAM SLAM algorithm based on LOAM [23,27]. optimized LOAM SLAM was adopted over GraphSLAM algorithms, as the latter are generally more computationally expensive, such that they consider all pose estimates in the calculation process. By contrast, EKF-based SLAM methods consider the last pose only [40]. 

The improved LOAM SLAM consists of three sequentially performed steps: key point extraction, LiDAR odometry, and LiDAR mapping. In the first stage, the critical points are categorized based on their curvature as either edges or planar points. Planar points correlate to the smallest curvature, whereas edge points correspond to the highest curvature. In the LiDAR odometry stage, a modified version of the iterative closest point matching algorithm is employed to reconstruct the LiDAR motion between two consecutive frames. Depending on the feature type, the distance between a point and a line or a point and a plane can be calculated. Finally, the recovered motion is refined by projecting the current frame onto the existing map and comparing it to it. More details about the LOAM SLAM can be found in [23].

Optimized LOAM SLAM is an enhanced variant of LOAM. The execution time is shortened due to the utilization of C++ libraries and tools optimized for improved computational speed. In addition, the method is not dependent on hard-coded parameters and can be run on either robot operating system (ROS) or LiDARView software. In addition, it is more generalized so that it operates on multiple LiDAR sensors, including Velodyne sensors. However, it is essential to highlight that optimized LOAM SLAM neither currently conducts loop closure nor it incorporates IMU measurements in the scan matching of successive frames. In this research, the optimized LOAM SLAM algorithm was executed on ROS. Two Python nodes (pc saver and traj saver) were created to subscribe to the trajectory and point cloud topics and preserve their respective outputs. Figure 6 and Figure 7 [41] show the LiDAR SLAM trajectory and the generated point cloud, respectively. 

### 2.3. Data Synchronization

All collected data were synchronized through GPS time stamp and resampled to 10 Hz to ensure correct data correspondence which is essential for subsequent processing stages. As such, the IMU data were downsampled from 200 Hz to 10 Hz for the raw accelerometers and gyroscopes measurements. Conversely, the camera pose estimations produced by Pix4D Mapper were up-sampled from approximately 0.5 Hz to 10 Hz. The resampling was accomplished using linear interpolation. 

Subsequently, coordinates transformation was performed to have all pose estimation in the same reference frame. The position and rotation transformations were executed using a homogenous transformation using 4 × 4 transformation matrices, which is computationally efficient. The LiDAR pose estimates were transformed from the local frame of the first point cloud of the Velodyne sensor stream into the WGS84 reference frame. Figure 8 depicts a graphical illustration of the sequence of transformations. 

Let $P^{Li}$ denotes a point captured in the local frame of the LiDAR and let $P^{ecef}$ denotes the same point expressed in the WGS84 reference frame. The sequence of homogenous transformations is presented by Equations (1) and (2).
(1)$$P^{ecef}={\left(R/t\right)}_{l}^{ecef}{\left(R/t\right)}_{b}^{l}{\left(R/t\right)}_{Li}^{b}P^{Li}$$
(2)$$R_{Li}^{l}=R_{b}^{l}R_{Li}^{b}$$
where ${\left(R/t\right)}_{l}^{ecef}$ is the homogenous transformation matrix from the *l*-frame to the WGS84 frame, ${\left(R/t\right)}_{b}^{l}$ is the homogenous transformation matrix from the *b*-frame to the *l*-frame, ${\left(R/t\right)}_{Li}^{b}$ is the homogenous transformation matrix from the LiDAR frame to the *b*-frame, $R_{Li}^{l}$ is the rotation matrix from the LiDAR frame to the *l*-frame, and $R_{Li}^{b}$ is the rotation matrix from the LiDAR frame to the *b*-frame.

## 3. System Architecture and Mathematical Models

A loosely coupled integration between the INS and the LiDAR SLAM is implemented using an EKF, which results in an integrated navigation solution. As shown in Figure 9, the raw IMU measurements are fed into a full IMU mechanization, which produces the position, velocity, and attitude angles of the inertial navigation system. The LiDAR scans are used as input to the optimized LOAM SLAM algorithm, which results in the position and attitude of the UAV. These are used as the measurement update to the IMU mechanization output during the update stage of the EKF. The result is the integrated INS/LiDAR navigation solution. The updated errors are fed back into the IMU mechanization, which forms a closed-loop error scheme. Two case studies were considered. The first one is a complete absence of the GNSS signal throughout the full UAV trajectory. In this case, the measurement update is from the LiDAR SLAM only. The second case study, on the other hand, includes updates from the GNSS PPP solution every 20 s. 

The EKF system model, measurement model, and other mathematical and stochastic equations are similar to [33]. In the prediction stage, the state vector is defined by Equation (3). Subsequently, in the update stage, the measurement update vector is given by Equation (4).
(3)$$\delta x={\left[\begin{array}{ccccc} \delta r & \delta v & \delta \varepsilon & \delta b_{a} & \delta b_{g} \end{array}\right]}^{T}$$
(4)$$\delta Z_{k}={\left[\begin{array}{cc} \delta r & \delta \varepsilon \end{array}\right]}^{T}={\left[\begin{array}{cccccc} \varphi_{imu-Li} & \lambda_{imu-Li} & h_{imu-Li} & p_{imu-Li} & r_{imu-Li} & y_{imu-Li} \end{array}\right]}^{T}$$
where $\delta r={\left[\begin{array}{ccc} \delta \phi & \delta \lambda & \delta h \end{array}\right]}^{T}$ is the position error vector; $\delta v={\left[\begin{array}{ccc} \delta v_{e} & \delta v_{n} & \delta v_{u} \end{array}\right]}^{T}$ is the velocity error vector, $\delta \varepsilon ={\left[\begin{array}{ccc} \delta p & \delta r & \delta y \end{array}\right]}^{T}$ is the attitude angles’ error vector; $\delta b_{a}={\left[\begin{array}{ccc} \delta b_{ax} & \delta b_{ay} & \delta b_{az} \end{array}\right]}^{T}$ is the accelerometer bias error vector; $\delta b_{g}={\left[\begin{array}{ccc} \delta b_{gx} & \delta b_{gy} & \delta b_{gz} \end{array}\right]}^{T}$ is the gyroscope bias error vector; $\phi_{imu-Li}$, $\lambda_{imu-Li}$, $h_{imu-Li}$ are the measurement errors for the geodetic latitude, longitude and height; and $p_{imu-Li}$, $r_{imu-Li}$, $y_{imu-Li}$ are the measurement errors for the pitch, roll, and yaw angles.

## 4. Results and Analysis 

### 4.1. First Case Study—Complete GNSS Outage

In this case study, a full GNSS signal outage is assumed along the whole trajectory of the UAV. Three navigation solutions were considered in this case study, namely, INS-only, LiDAR SLAM-only, and the integrated INS/LiDAR SLAM solution. The INS-only solution represents the solution obtained through IMU mechanization, while the LiDAR SLAM-only solution is the one produced using the optimized LOAM SLAM. Finally, the third solution is the integrated INS/LiDAR SLAM solution using EKF. The position and attitude of the three navigation solutions were compared to the ground truth, which is the interpolated camera solution. Figure 10 shows the position errors in the ENU local reference frame, while Figure 11 illustrates the errors of the attitude angles (roll, pitch, and yaw) for three navigation solutions. The error statistics of the position and attitude are shown in Table 2 for the three navigation solutions.

Figure 10 shows that the INS solution drifts significantly over time, indicating that it cannot be used solely for accurate navigation. However, in comparison with the INS solution, the integrated navigation system produced significantly less errors. Therefore, the integrated system was tuned to increase the weights of LiDAR SLAM position updates, while lowering those of the INS solution. As a result, the combined INS/LiDAR position solution was somewhat similar to the LiDAR SLAM solution. As shown in Figure 10 and Table 2, the up-direction error is significantly larger than the horizontal counterpart. The reason for this behavior is due to the nature of the collected point clouds, such that the distribution of the points is only in one direction (below the lidar sensor), which causes poor vertical geometry. Similar behavior was presented in [33], where the estimations for the up-direction from the LiDAR SLAM was sensitive to the geometry of the detected features. 

In contrast, the integrated INS/LiDAR solution in Figure 11 was tuned to follow the INS solution for pitch and roll angles, which outperformed those of the LiDAR SLAM. However, the integrated system followed the yaw angle produced by the LiDAR SLAM, which was more accurate than that of the INS solution. It is worth mentioning that the IMU measures the vehicle’s accelerations and angular rotations directly, after which the attitude was estimated, which leads to more accurate estimations for the attitude in comparison to position estimation. However, when the drone performs sharp turns, the yaw angles estimations from the IMU mechanization drifts significantly, in which case, the LiDAR SLAM estimates are better for the yaw. This behavior echoes with [33] for the pitch and roll angles, albeit not the yaw angle. The reason for this is due the different nature of the datasets, terrestrial in [33] versus UAV system in this paper. The yaw angle estimate from the IMU mechanization in case of UAV navigation will be more sensitive to turns because the UAV turns are typically drastically sharper than those of cars. 

The RMSE of the integrated INS/LiDAR position errors and the yaw angle converged towards that of the LiDAR SLAM, while the RMSE of the pitch and roll angles were closer to that of the mechanization errors. In Figure 12, LiDAR SLAM and INS/LiDAR trajectories are compared to the ground truth for the first case study (complete GNSS signal outage).

### 4.2. Second Case Study—GNSS PPP Assistance

In the second case study, updates from the GNSS PPP solution were considered to improve the LiDAR SLAM results, and thereby fill in the GNSS gaps, as mentioned in Section 2. The GNSS position updates are fed into the EKF every 20 s, whenever they are available, while the filter continued to receive attitude updates from the LiDAR SLAM.

The same approach for the filter tuning was adopted as discussed in the first case study. As a result, a similar performance can be observed for the position and attitude errors of the integrated system (Figure 13 and Figure 14). However, as shown in Table 3, the errors are significantly reduced in comparison with the previous case study, which is attributed to the GNSS updates. A comparison between LiDAR SLAM and GNSS/INS/LiDAR trajectories with respect to the ground truth for the second case study is shown in Figure 15. As shown in Table 4, it is noticeable that the GNSS updates led to a lower RMSE of approximately 51% and 78% in the horizontal and up directions, respectively. Additionally, the reductions in the RMSE of the roll and yaw angles were roughly 13% and 28%, respectively. However, there existed no significant change in the pitch angle. 

It is worth mentioning that the positioning accuracy for this case study mainly depends on that of the PPP solution. As mentioned in Section 2.2, the accuracy of the GNSS PPP solution was in the dm-m level accuracy. Should a higher precision PPP solution be available, the integrated solution will follow.

## 5. Conclusions

In this study, a GNSS/INS/LiDAR integrated navigation system was developed to overcome numerous and prolonged GNSS signal outages caused by a malfunctioning GNSS antenna during data collection. To recover the full trajectory of the UAV, the LiDAR data were processed using the optimized LOAM SLAM algorithm, which yielded position and rotation estimations of the sensor. The camera images were processed with the help of four corner GCPs using Pix4D Mapper software, which resulted in precise camera poses that served as ground truth. All collected sensor data were timestamped by the GPS time, and therefore all datasets were synchronized to match the same frequency of the LiDAR data. Two case studies were considered. In the first case study, a LC integration between the INS and the LiDAR SLAM solution was implemented using an EKF. The positioning accuracy of this case study depends mainly on the LiDAR SLAM accuracy, which yielded a larger error in the up direction in comparison with the horizontal direction. This is essentially caused by the poor vertical geometry. In contrast, the integrated INS/LiDAR solution was tuned to follow the INS solution for pitch and roll angles, which outperformed those of the LiDAR SLAM. However, the integrated system followed the yaw angle produced by the LiDAR SLAM, which was more accurate than that of the INS solution.

A similar integration was executed in the second case study, albeit with updates from the GNSS PPP solution every about 20 s. The same approach for filter tuning was adopted as in the first case study. It was shown that significant improvement in the latter case is obtained in the pose estimation compared to the full absence of GNSS. The positioning RMSE was reduced by 51% and 78% in the horizontal and up directions, respectively, while the RMSE of the roll and yaw angles was reduced by 13% and 28%, respectively. However, the RMSE of the pitch angle was increased by about 13%.

## Figures and Tables

**Figure 1 sensors-22-09908-f001:**
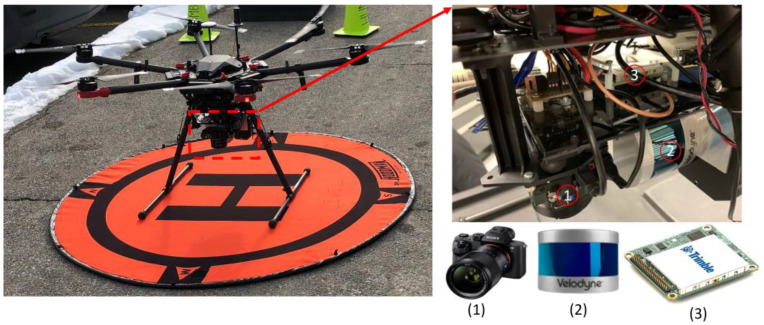
UAS used in the study and its payload: (1) Sony α7R II camera (2) Velodyne PUCK LiDAR sensor (3) Applanix APX-15 GNSS/IMU board.

**Figure 2 sensors-22-09908-f002:**
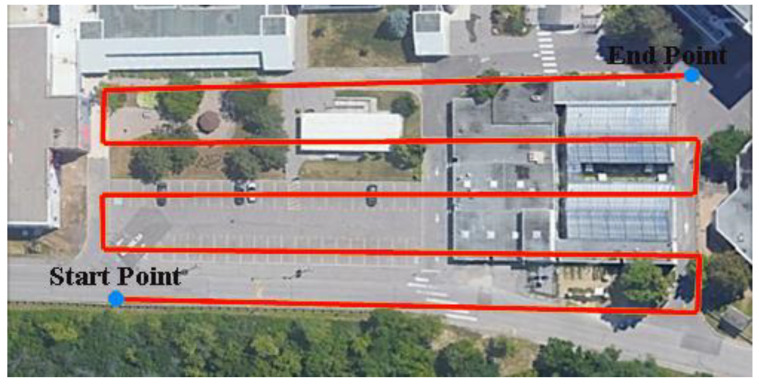
UAS flight lines above the study area.

**Figure 3 sensors-22-09908-f003:**
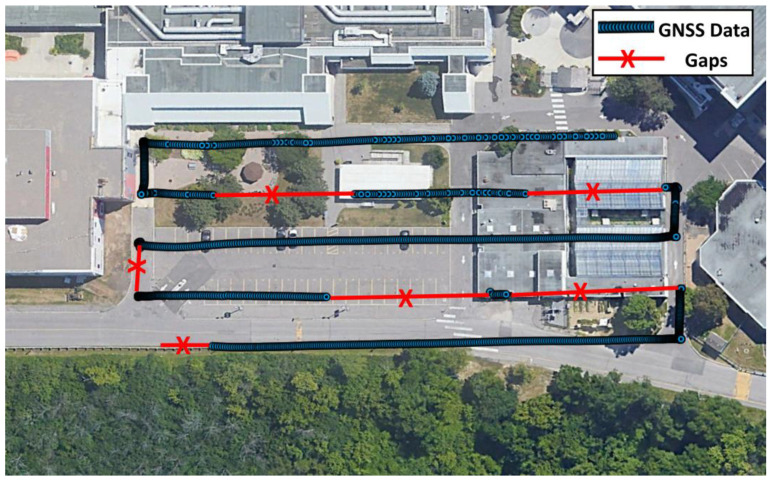
GNSS data gaps due to an unexpected antenna malfunction.

**Figure 4 sensors-22-09908-f004:**
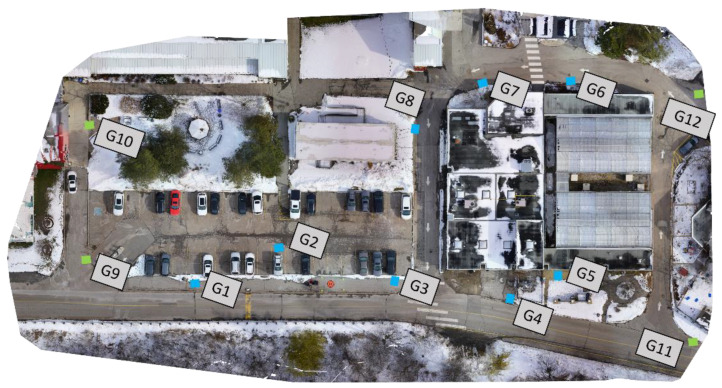
Ground control points (GCPs) and checkpoints distribution in study area.

**Figure 5 sensors-22-09908-f005:**
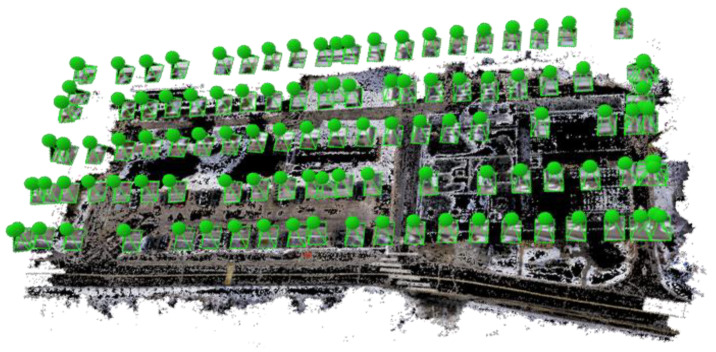
Camera poses estimated using Pix4D mapper software.

**Figure 6 sensors-22-09908-f006:**
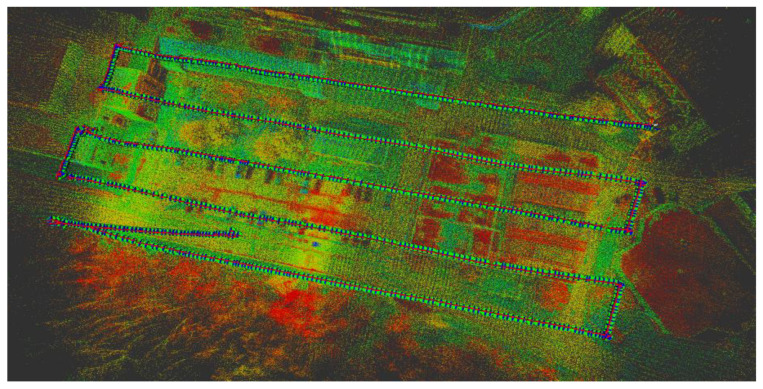
LiDAR SLAM trajectory generated using optimized LOAM SLAM algorithm [41].

**Figure 7 sensors-22-09908-f007:**
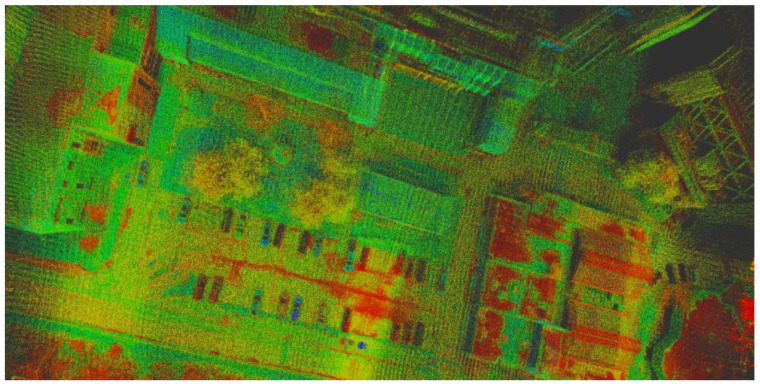
LiDAR point cloud generated using optimized LOAM SLAM algorithm [41].

**Figure 8 sensors-22-09908-f008:**
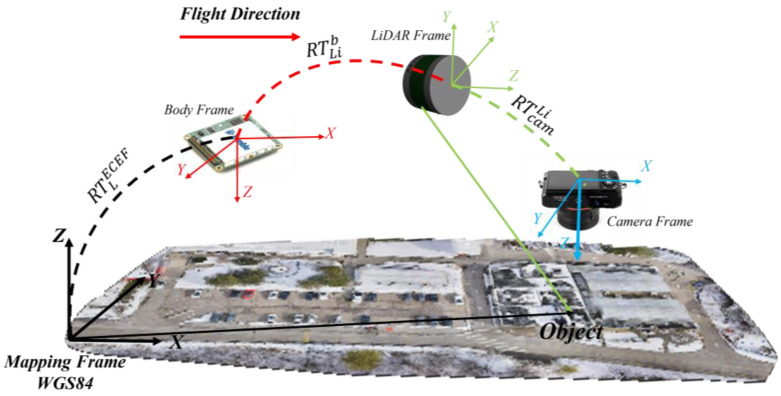
Graphical illustration of the data collection platform extrinsic transformation.

**Figure 9 sensors-22-09908-f009:**
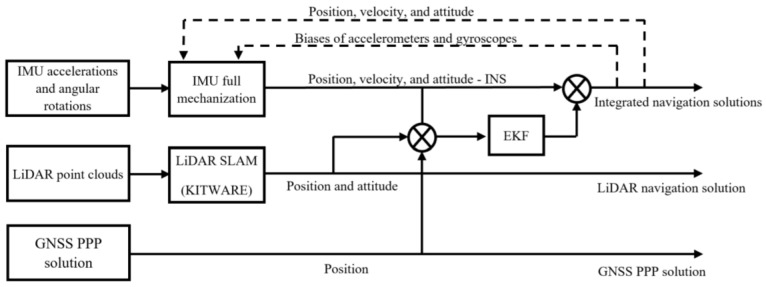
A block diagram for the GNSS/INS/LiDAR SLAM LC integration.

**Figure 10 sensors-22-09908-f010:**
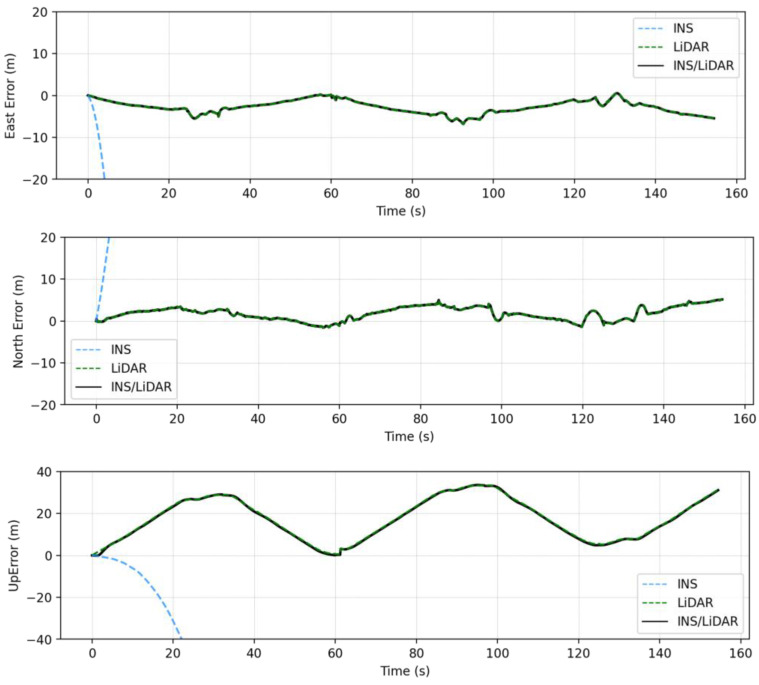
Complete GNSS signal outage: position errors (ENU).

**Figure 11 sensors-22-09908-f011:**
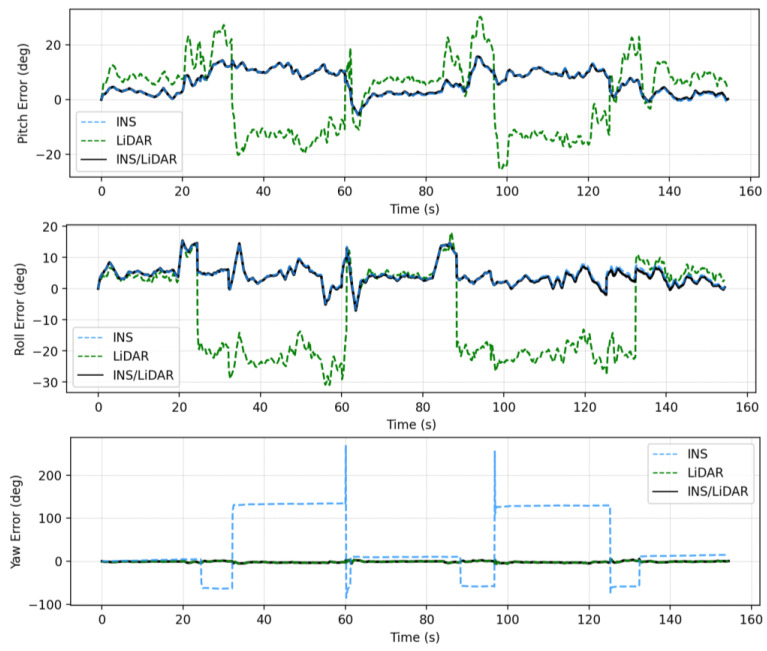
Complete GNNS signal outage: errors of attitude angles (roll, pitch, and yaw).

**Figure 12 sensors-22-09908-f012:**
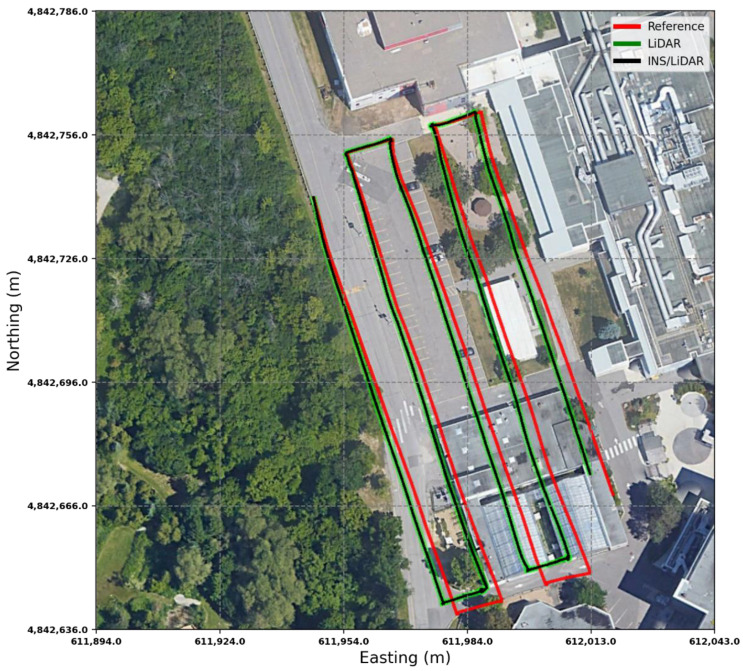
Complete GNNS signal outage: comparison of trajectories.

**Figure 13 sensors-22-09908-f013:**
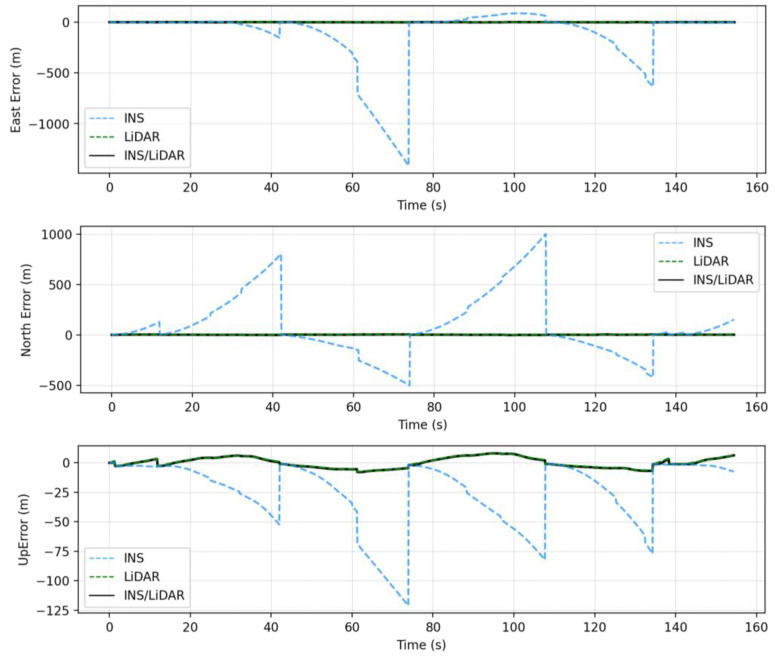
GNSS assisted: position errors (ENU).

**Figure 14 sensors-22-09908-f014:**
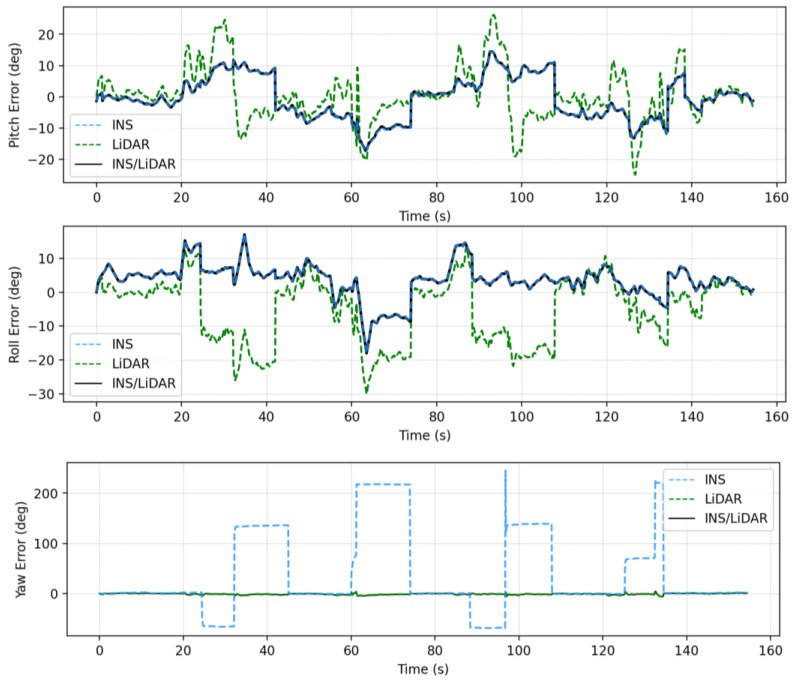
GNSS assisted: errors of attitude angles (roll, pitch, and yaw).

**Figure 15 sensors-22-09908-f015:**
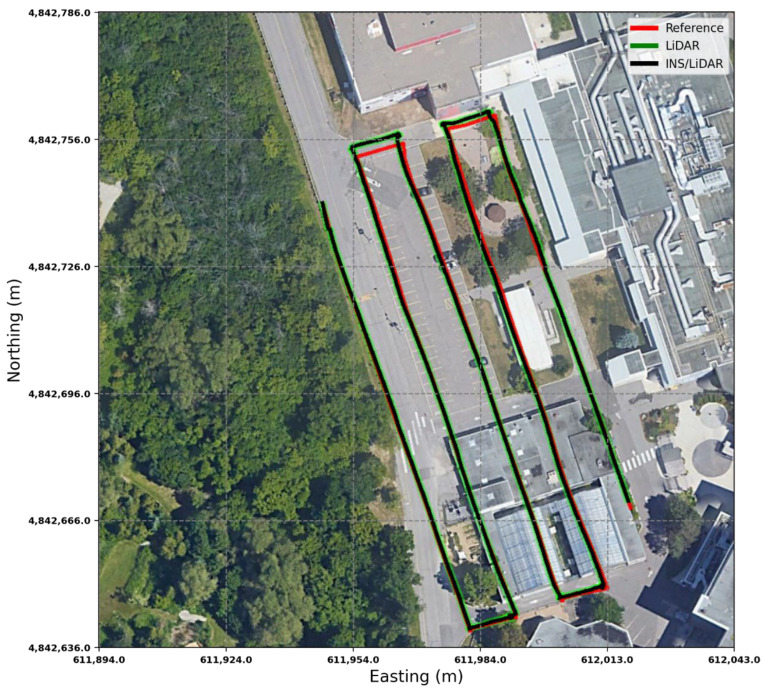
GNSS assisted: comparison of trajectories.

**Table 1 sensors-22-09908-t001:** Absolute camera position (m) and orientation (deg) error statistics.

	X	Y	Z	Omega	Phi	Kappa
Mean	0.003	0.004	0.003	0.008	0.007	0.003
Sigma	0.001	0.001	0.001	0.002	0.001	0.001

**Table 2 sensors-22-09908-t002:** Complete GNNS signal outage—position (m) and attitude (deg) error statistics.

	INS			LiDAR			INS/LiDAR		
	Mean	RMSE	Max	Mean	RMSE	Max	Mean	RMSE	Max
East	−8708.58	11,610.52	25,045.04	−2.75	3.16	6.79	−2.75	3.16	6.79
North	147.08	436.67	894.45	1.77	2.37	5.18	1.77	2.37	5.18
Horizontal	8743.97	11,618.73	25,061.01	3.49	3.95	7.52	3.49	3.95	7.52
Up	−789.50	1056.81	2292.92	17.42	19.95	33.43	17.15	19.78	33.60
Pitch	4.54	5.53	15.27	−8.44	16.19	31.00	4.27	5.37	15.43
Roll	6.17	7.67	15.47	1.50	12.89	30.29	6.24	7.65	15.72
Yaw	42.56	83.37	269.91	−1.33	2.11	5.94	−1.36	2.17	5.98

**Table 3 sensors-22-09908-t003:** GNSS assisted—position (m) and attitude (deg) error statistics.

	INS			LiDAR			INS/LiDAR		
	Mean	RMSE	Max	Mean	RMSE	Max	Mean	RMSE	Max
East	−76.11	198.80	612.58	−0.69	0.94	2.71	−0.69	0.94	2.71
North	106.86	151.63	405.55	1.09	1.70	4.70	1.09	1.70	4.70
Horizontal	175.18	250.03	665.13	1.69	1.94	5.22	1.70	1.94	5.22
Up	−5.78	12.61	38.42	0.27	4.33	7.82	0.06	4.38	8.00
Pitch	0.19	4.33	19.18	−5.15	10.84	30.09	3.55	6.06	18.05
Roll	1.19	4.58	12.46	0.29	8.39	26.17	−0.53	6.69	17.31
Yaw	39.13	90.44	245.15	−0.83	1.52	6.18	−0.84	1.56	6.20

**Table 4 sensors-22-09908-t004:** Performance comparison between the two study cases (RMSE improvement).

	Case 1	Case 2	Accuracy Improvements (%)
East	3.16	0.94	70.34
North	2.37	1.70	28.26
Horizontal	3.95	1.94	50.85
Up	19.78	4.38	77.85
Pitch	5.37	6.06	−12.82
Roll	7.65	6.69	12.61
Yaw	2.17	1.56	28.11

## Data Availability

The data presented in this study are not publicly available.

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
