# Peer review of "A GNSS/INS/LiDAR Integration Scheme for UAV-Based Navigation in GNSS-Challenging Environments"

_sensors, 2022, doi:10.3390/s22249908_

Round 1
Reviewer 1 Report
The most challenging aspect of UAV navigation is to maintain accurate and reliable pose estimation. GNSS or GNSS/INS systems are often used for UAV positioning in outdoor environments. However, if the receiver antenna is blocked, the GNSS signal will be interrupted, which will lead to serious pose drift of GNSS/INS. In this paper, a GNSS/INS/LiDAR integrated SLAM navigation system based on UAV is studied, and the proposed method is to overcome the UAV pose drift caused by frequent and long-term GNSS signal interruption, which has great significance. The research of the thesis is good on the whole, but the experimental effect is not very good, and the paper still has room for improvement .
Comments:
2.2. Data Preprocessing:"The GNSS data was then processed in precise point positioning(PPP) mode, which was successful, but the obtained results were at dm-m level accuracy with many gaps." Actually this is mainly due to the unmodeled errors, one may point out this issue better
A composite stochastic model considering the terrain topography for real-time GNSS monitoring in canyon environments, Journal of Geodesy, 2022, 96: 79.
Code and phase multipath mitigation by using the observation-domain parameterization and its application in five-frequency GNSS ambiguity resolution. GPS Solutions, 2021, 25: 144.
2.2. Data Preprocessing:"In the LiDAR odometry stage, the iterative closest point matching algorithm is employed to reconstruct the LiDAR motion between two consecutive frames." LOAM does not use the traditional iterative closest point matching algorithm. Please describe it clearly .
2.2. Data Preprocessing:As mentioned in the last few sentences, optimized LOAM SLAM is independent of the ROS. However, it goes on to say that "two Python nodes (pc saver and traj saver) were created to subscribe to the trajectory and point cloud topics and retain their respective outputs.," Is there any contradiction in using ROS?
4.1. First Case Study—Complete GNSS Outage:As can be seen from Figure 10-11 and Table 2, the algorithm drifts up to several meters, up to 30m at altitude, and the error is still large with GNSS signals behind. As far as I know, the classical LOAM algorithm will not have such a large error in such a large-scale scene. Please explain the reason for such a large drift.
4.3. Second Case Study — GNSS PPP Assistance:The serial number of chapter title is marked incorrectly, and the chapter 4.2 is missing.
6.Conclusions:The same problem, the chapter title number mark error, this should be the fifth section.
6.Conclusions:in conclusion, we should analyze that result in depth around the core content of the paper, not just show the experimental result.
Reviewer 2 Report
Comments for Manuscript No.2058315
Dear Sir/Madam,
The paper entitled “A GNSS/INS/LiDAR Integration Scheme for UAV-based Navigation in GNSS-Challenging Environments” submitted by Ahmed Elamin et al. presented an integrated UAV navigation system to resolve the problem of GNSS signal outages. The paper is well written and very interesting to read.
However, i found some serious faults that they should be revised before publishing this paper. My recommendations for paper revision are written on a separate page.
Best regards,
Reviewer.
Common comments:
References
1. Many articles listed in references are not cited in text.
ex) Reference no. from 39 to 43, and 45.
2. Journal names are missed in most of references.
ex) 1, 3, 6, 10, 12 and so on.
Abstract (the last sentence)
Additionally, the RMSE of the roll and yaw angles was reduced by 13% and 30%, respectively. It should be revised as follow:
Additionally, the RMSE of the pitch and yaw angles was reduced by 13% and 30%, respectively. Meanwhile, the RMSE of roll angle was increased by 13%.
See details in 4.3.
2.1 Data Acquisition
1. Why did you design the flight height so low at 25m? Although it is difficult to receive the GNSS signal due to tall buildings nearby.
2. How long the lines in flight direction and cross direction?
3. What is the total weight of the entire integrated system?
2.2 Data Preprocessing
1. Why did you design the flight height so low at 25m? Although it is difficult to receive the GNSS signal due to tall buildings nearby.
2. In general, GNSS positioning accuracy in a differential mode affected by the distance from rover antenna to ground base antenna. Where did you install the GNSS base station in ground, and how far is it from study area?
3. How much large the variation of flight attitude due to mass imbalance of a UAV?
2.3 Data Synchronization
1. How can you get accurate camera pose data using up-sampling from 0.5 Hz to 10 Hz with simple linear interpolation? Because it is impossible to calculate exact pose data due to unstable drone flight during short time.
3. System Architecture
1. Why did you set 20 seconds as update interval for GNSS PPP solution? I mean this interval is too long to bridge gaps of GNSS signal.
2. Check the correctness of position and attitude by GNSS PPP solution in Fig.9.
How can we get the attitude from standalone GNSS?
4.1 First Case Study
1. According to Fig. 10, the errors in E and N are very accurately inversely correlated, and the errors in U has a regular pattern except INS data, that is, sine curve.
According to Table 2, the position RMSEs in U direction of Lidar and INS/Lidar too much large in comparison to errors in E and N.
I would like to know the reasons for above results.
2. According to Table 2 and Fig. 10, the RMSE in yaw angle is too large compare with roll and pitch. This result is not reasonable; because you used the same sensors during all UAV flight, the accuracy result must be about the same. The Figure 10 shows clearly that the RMSE of yaw is too much large in specific flight lines.
3. As can be seen in Figure 11, since the value and characteristics of RMSE are significantly different for each flight strip, accuracy analysis must be performed separately for each strip.
4.3 Second Case Study
1. The sub-chapter should be corrected as 4.2 Second Case Study.
2. As mentioned earlier, if the GNSS position is updated every 20 seconds, flight lines 2 and 4 may have no GNSS information at all, as shown in Figure 3.
3. Check the accuracy improvement by comparing Tables 2 and 3 in detail, accuracy improvement calculated from the change of RMSE of attitude angles of INS/ LiDAR calculated as follows:
|
Attitude angle |
RMSE INS/Lidar No update/with update |
Examples of Calculated Value |
Accuracy Improvement (%) |
|
Role |
5.37/6.06 |
(5.37-6.06)/5.37 |
-13 |
|
Pitch |
7.65/6.69 |
(7.65-6.69)/7.65 |
13% |
|
Yaw |
2.17/1.56 |
(2.17-1.56)/2.17 |
28% |
- The End -
Round 2
Reviewer 2 Report
Dear Authors,
Thank you for your detailed revision and kind explanation.
But some critical issues are remaind unresoved as follows:
Figure 11: the error of yaw angles
You explained: the reason why yaw error is greater than that of roll or pitch is attributed to the sharp turn. Even though the same drone turns in the same pattern in all flights, as shown in Figure 11, the yaw error only increases in the 2nd and 4th strips. In addition, Even in the case of GNSS assisted, shown in Figure 14, a yaw error of about 180 degrees occurs only in strips 2 and 4. This time, only the sign of the error changed from + to -, conversely to Fig.11. It is believed that this error was caused by the orientation error of the reversed coordinates after the drone rotated 180 degrees.
Figure 13 and Figure 14:
The size of the horizontal axis should be 10 times the current value, i.e. the last number should be 1600.
Conclusions
Please revise the last sentence about pitch error like the last sentence in abstract.
-The End-
